# The *Drosophila Baramicin* polypeptide gene protects against fungal infection

**Mark Austin Hanson**[1]*, **Lianne B. Cohen**[2], **Alice Marra**[1], **Igor Iatsenko**[1,3], **Steven A. Wasserman**[2], **Bruno Lemaitre**[1]*

**1** Global Health Institute, School of Life Science, École Polytechnique Fédérale de Lausanne (EPFL), Lausanne, Switzerland, **2** Division of Biological Sciences, University of California San Diego (UCSD), La Jolla, California, United States of America, **3** Max Planck Institute for Infection Biology, Berlin, Germany

\* mark.hanson@epfl.ch (MAH); bruno.lemaitre@epfl.ch (BL)

## Abstract

The fruit fly *Drosophila melanogaster* combats microbial infection by producing a battery of effector peptides that are secreted into the haemolymph. Technical difficulties prevented the investigation of these short effector genes until the recent advent of the CRISPR/CAS era. As a consequence, many putative immune effectors remain to be formally described, and exactly how each of these effectors contribute to survival is not well characterized. Here we describe a novel *Drosophila* antifungal peptide gene that we name *Baramicin A*. We show that *BaraA* encodes a precursor protein cleaved into multiple peptides via furin cleavage sites. *BaraA* is strongly immune-induced in the fat body downstream of the Toll pathway, but also exhibits expression in other tissues. Importantly, we show that flies lacking *BaraA* are viable but susceptible to the entomopathogenic fungus *Beauveria bassiana*. Consistent with *BaraA* being directly antimicrobial, overexpression of *BaraA* promotes resistance to fungi and the IM10-like peptides produced by *BaraA* synergistically inhibit growth of fungi in vitro when combined with a membrane-disrupting antifungal. Surprisingly, *BaraA* mutant males but not females display an erect wing phenotype upon infection. Here, we characterize a new antifungal immune effector downstream of Toll signalling, and show it is a key contributor to the *Drosophila* antimicrobial response.

OPEN ACCESS

**Data Availability Statement:** All relevant data are within the manuscript and its Supporting information files.

## Author summary

The ways that animals combat infection involve complex molecular pathways that are triggered upon microbial challenge. While a great deal is known about which pathways are key to a successful defence response, far less is known about exactly what elements of that response are critical to combat a given infection. Using the fruit fly–a genetic workhorse of Biology–we recently showed that a class of host-encoded antibiotics called "antimicrobial peptides" are essential for defence against bacterial infection, but do not contribute as strongly to defence against fungi. However a number of fly immune peptides remain uncharacterized, possibly explaining this gap in our understanding of the fly antifungal defence. Here we describe a novel antifungal peptide gene of fruit flies, and show that it is a major contributor to the fly antifungal defence response. We also found that

**Funding:** We would like to acknowledge the following sources of funding: BL: recipient of Swiss National Science Foundation, Sinergia grant number CRSII5_186397 http://www.snf.ch/en/funding/programmes/sinergia/Pages/default.aspx BL: recipient of Novartis Foundation grant number 532114 https://www.novartis.com/our-focus/healthcare-professionals/novartis-external-funding SW - recipient of NIH R01 grant number GM050545 https://grants.nih.gov/grants/funding/r01.htm The funders had no role in study design, data collection and analysis, decision to publish, or preparation of the manuscript.

**Competing interests:** The authors have declared that no competing interests exist.

this gene seems to regulate a behaviour that flies perform after infection, paralleling exciting recent findings that these genes are involved in neurological processes. Collectively, we clarify a key part of the fly antifungal defence, and contribute an important piece to help explain the logical organization of the immune defence against microbial infection.

## Introduction

The innate immune response provides the first line of defence against pathogenic infection. This reaction is usually divided into three stages: i) the recognition of pathogens through dedicated pattern recognition receptors, ii) the activation of conserved immune signalling pathways and iii) the production of immune effectors that target invading pathogens [1,2]. The study of invertebrate immune systems has led to key observations of broad relevance, such as the discovery of phagocytosis [3], antimicrobial peptides (AMPs) [4], and the implication of Toll receptors in metazoan immunity [5]. Elucidating immune mechanisms, genes, and signalling pathways has greatly benefited from investigations in the fruit fly *Drosophila melanogaster*, which boasts a large suite of molecular and genetic tools for manipulating the system. One of the best-characterized immune reactions of *Drosophila* is the systemic immune response. This reaction involves the fat body (an analog of the mammalian liver) producing immune effectors that are secreted into the haemolymph. In *Drosophila*, two NF-κB signalling pathways, the Toll and Imd pathways, regulate most inducible immune effectors: the Toll pathway is predominantly activated in response to infection by Gram-positive bacteria and fungi [5,6], while the immune-deficiency pathway (Imd) responds to the DAP-type peptidoglycan most commonly found in Gram-negative bacteria and a subset of Gram-positive bacteria [7]. These two signalling pathways regulate a transcriptional program that results in the massive synthesis and secretion of humoral effector peptides [6,8]. Accordingly, mutations affecting the Toll and Imd pathways cause extreme susceptibilities to systemic infection that reflect the important contribution of these pathways to host defence. The best-characterized immune effectors downstream of these pathways are antimicrobial peptides (AMPs). AMPs are small and often cationic peptides that disrupt the membranes of microbes, although some have more specific mechanisms [9]. Multiple AMP genes belonging to seven well-characterized families are induced upon systemic infection [10]. However transcriptomic analyses have revealed that the systemic immune response encompasses far more than just the canonical AMPs. Many uncharacterized genes encoding small secreted peptides are induced to high levels downstream of the Toll and Imd pathways, pointing to the role for these peptides as immune effectors [11]. In parallel, MALDI-TOF analyses of the haemolymph of infected flies revealed the induction of 24 peaks–mostly corresponding to uncharacterized peptides–that were named "IMs" for Immune-induced Molecules (IM1-IM24) [8]. Many of the genes that encode these components of the immune peptidic secretome had remained unexplored until recently. This is mainly due to the fact that these IMs belong to large gene families of small genes that were not typically disrupted using random mutagenesis [10,12].

The CRISPR/Cas9 gene editing approach now allows the necessary precision to delete small genes, singly or in groups, providing the opportunity to dissect effector peptide functions. In 2015 a family of 12 related IM-encoding genes, unified under the name *Bomanins*, were shown to function downstream of Toll. Importantly, a deletion removing 10 out of the 12 Bomanins revealed their potent contribution to defence against both Gram-positive bacteria and fungi [13]. While Bomanins contribute significantly to Toll-mediated defence, their molecular functions are still unknown and it is unclear if they are directly antimicrobial [14]. Two other IMs

encoding IM4 and IM14 (renamed *Daisho1* and *Daisho2*, respectively) were shown to contribute downstream of Toll to resistance against *Fusarium* fungi. Interestingly, Daisho peptides bind to fungal hyphae, suggesting direct antifungal activity [15]. Finally a systematic knockout analysis of *Drosophila* AMPs revealed that they play an important role in defence against Gram-negative bacteria and some fungi, but surprisingly little against Gram-positive bacteria [16]. An unforeseen finding from these recent studies is the high degree of AMP-pathogen specificity: this is perhaps best illustrated by the specific requirement for *Diptericin*, but not other AMPs, in defence against *Providencia rettgeri* [16,17]. Collectively, these studies in *Drosophila* reveal that immune effectors can be broad or specific in mediating host-pathogen interactions. Understanding the logic of the *Drosophila* effector response will thus require a careful dissection of the remaining uncharacterized immune induced peptides.

Previous studies identified an uncharacterized Toll-regulated gene (*CG18279/CG33470*), which we rename "*BaraA*" (see below), that encodes several IMs, indicating a role in the humoral response. Here, we have improved the annotation of IMs produced by *BaraA* to include: IM10, IM12 (and its sub-peptide IM6), IM13 (and its sub-peptides IM5 and IM8), IM22, and IM24. Using a *BaraA* reporter, we show that *BaraA* is not only immune-induced in the fat body, but also expressed in the head, and nervous system tissue including the eyes, and ocelli. Importantly, we show that flies lacking *BaraA* are viable but susceptible to specific infections, notably by the entomopathogenic fungus *Beauveria bassiana*. Consistent with this, the IM10-like peptides produced by *BaraA* inhibit fungal growth in vitro when combined with the antifungal Pimaricin. Surprisingly, *BaraA* deficient males also display a striking erect wing behaviour upon infection. Collectively, we identify a new antifungal immune effector downstream of Toll signalling, improving our knowledge of the *Drosophila* antimicrobial response.

## Results

### *BaraA* is regulated by the Toll pathway

A previous microarray study from De Gregorio et al. [11] suggested that *BaraA (CG18279/ CG33470)* is primarily regulated by the Toll pathway, with a minor input from the Imd pathway (Fig 1A). Consistent with this, we found several putative NF-κB binding sites upstream of the *BaraA* gene (guided by previous reports [18–20]). Notably there are two putative binding sites for Relish, the transcription factor of the Imd pathway and three putative binding sites for the Dif/Dorsal transcription factors acting downstream of Toll (S1A Fig and S1). We challenged wild-type flies and Imd or Toll pathway mutants ($Rel^{E20}$ and $spz^{rm7}$ respectively) with the yeast *Candida albicans*, the Gram-negative bacterium *Escherichia coli*, or the Gram-positive bacterium *Micrococcus luteus*. RT-qPCR analysis confirms that *BaraA* is abolished in $spz^{rm7}$ flies similar to the Toll-regulated *BomBc3* gene (Fig 1B), but remains highly inducible in $Rel^{E20}$ flies (S1B Fig). Collectively, the expression pattern of *BaraA* is reminiscent of the antifungal peptide gene *Drosomycin* with a primary input by the Toll pathway and a minor input from the Imd pathway [10,21].

To further characterize the expression of *BaraA*, we generated a *BaraA-Gal4* transgene in which 1675bp of the *BaraA* promoter sequence is fused to the yeast transcription factor Gal4. Monitoring GFP in *BaraA-Gal4>UAS-mCD8-GFP* flies (referred to as *BaraA>mGFP*) confirms that the *BaraA* reporter is highly induced in the fat body after infection by *M. luteus*, but less so by *E. coli* (Fig 1C). This result is consistent with a recent time course study that found Toll-regulated genes (including *BaraA*) were rapidly induced after injection stimulating the Imd pathway, but this principally Imd-based induction resolves to nearly basal levels within 48 hours [22] (and see S1B and S1C Fig). Additionally, larvae pricked with *M. luteus* show a robust GFP signal primarily stemming from the fat body when examined 2hpi (S1D Fig). We

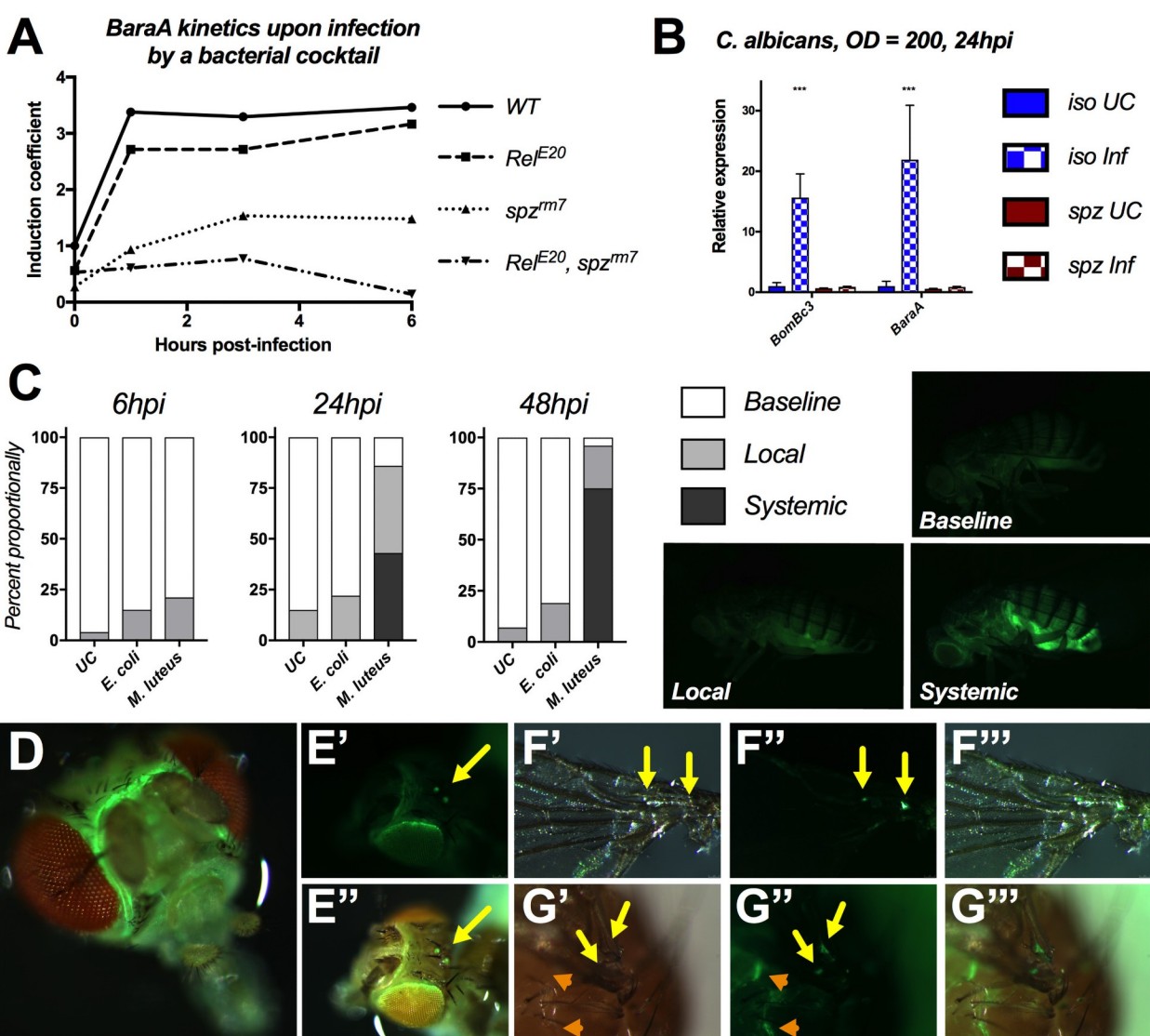

**Fig 1. *BaraA* is an immune-induced gene regulated by the Toll pathway. A)** Expression profile of *BaraA* upon bacterial challenge by a mixture of *E. coli* and *M. luteus* (from De Gregorio et al. [11]). Induction coefficient reports a $\text{Log}_{10}$-fold calculation then normalized to unchallenged wild-type expression levels (see De Gregorio et al. [11]). **B)** *BaraA* expression profiles in wild-type and *spz^rm7* flies upon septic injury with the yeast *C. albicans. BomBc3* is used as an inducible control gene for the Toll pathway. Significance relative to *iso-UC* indicated as *** = p < .001. Additional gene expression measurements are shown in S1B and S1C Fig. **C)** Use of a *BaraA* reporter reveals that *BaraA* induction upon infection is primarily driven by the fat body in adults, and results in a strong and systemic GFP signal upon pricking with OD = 200 *M. luteus* (stimulating the Toll pathway), but less so by *E. coli* (stimulating the Imd pathway) 24hpi and 48hpi ($\chi^2$ p < .001, N = 82). **D-G)** Baseline *BaraA>mGFP* is highly expressed in the head (**D**), at the border of the eyes and in the ocelli (**E**), in the wing veins (**F-G** yellow arrows), and beneath the cuticle in the thorax (**G**, orange arrowheads).

also observed a constitutive GFP signal in the headcase of adults (Fig 1D), including the border of the eyes and the ocelli (Fig 1E). Dissection confirmed that the *BaraA* reporter is expressed in brain tissue, including posterior to the central brain furrow in adults and at the posterior of the ventral nervous system in larvae. Other consistent signals include GFP in the wing veins and subcutaneously along borders of thoracic pleura in adults (Fig 1F and 1G), and in spermatheca of females (S1E Fig). There was also sporadic GFP signal in other tissues that included the larval hindgut, the dorsal abdomen of developing pupae, and the seminal vesicle of males.

These expression patterns largely agree with data reported in FlyAtlas1 (wherein *BaraA* is called "*IM10*") [23].

## *Baramicin A* encodes a precursor protein cleaved into multiple peptides

Previous studies using bioinformatics and proteomics have suggested that four highly immune-induced peptides (IM10, IM12, IM13, and IM24) are encoded in tandem as a single polypeptide precursor by *CG33470* (aka *IMPPP/BaraA*) [8,24]. Some less-abundant sub-peptides (IM5, IM6, and IM8) are also produced by additional cleavage or degradation of IM12 and IM13 [24]. Using a newly generated null mutant ("*ΔBaraA*," described below and design shown in Fig 2A), we analyzed haemolymph samples of wild-type and *ΔBaraA* flies infected with a bacterial mixture of *E. coli* and *M. luteus* by MALDI-TOF analysis. We confirmed the loss of the seven immune-induced peaks corresponding to IMs 5, 6, 8, 10, 12, 13, and 24 in *ΔBaraA* flies (Fig 2A). We also noticed that an additional immune-induced peak at ~5975 Da was absent in our *BaraA* mutants. Upon re-visiting the original studies that annotated the *Drosophila* IMs, we realized this peak corresponded to IM22, whose sequence was never determined [8,24] (see S1 Text and S2 Data for details). We subjected haemolymph from infected

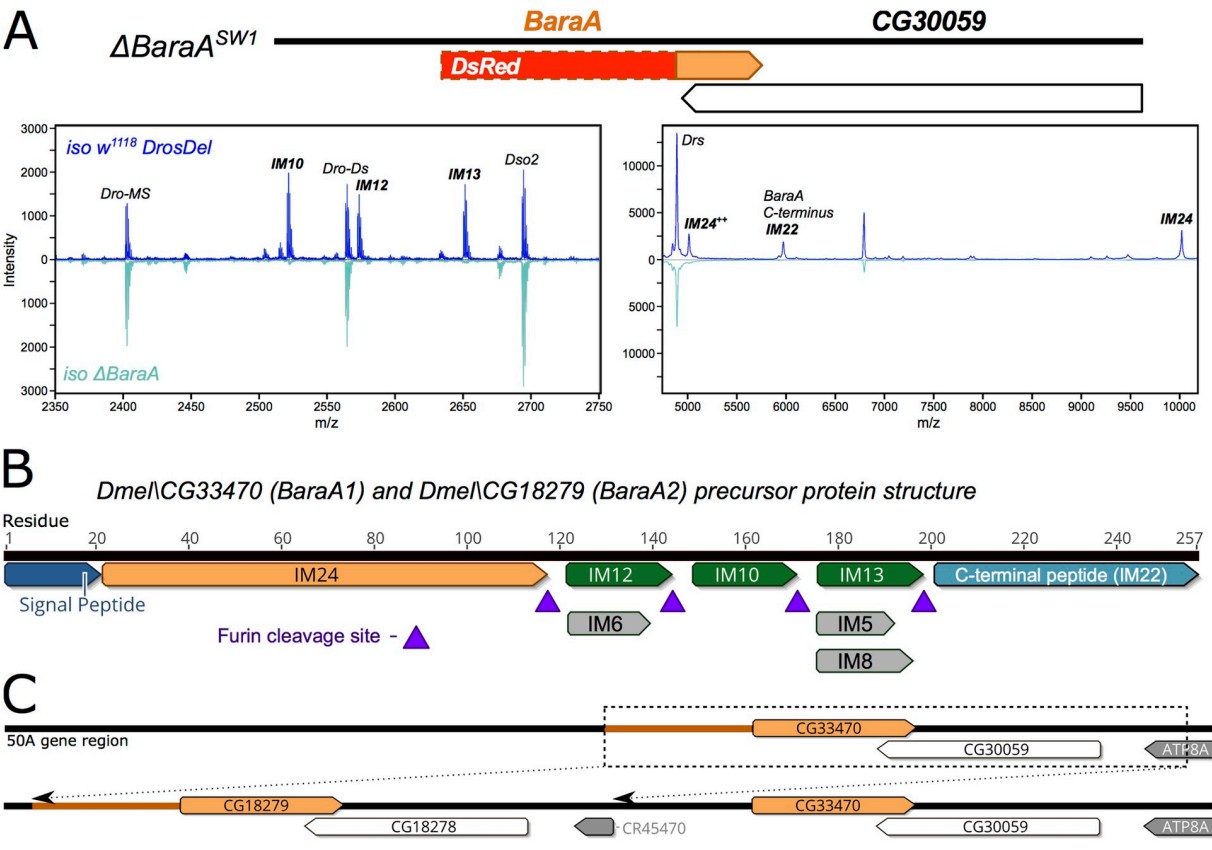

**Fig 2. The *BaraA* gene structure. A)** MALDI-TOF analysis of haemolymph from *iso w^1118* wild-type and *iso ΔBaraA* flies 24 hours post-infection (hpi) confirms that *BaraA* mutants fail to produce the IM10-like and IM24 peptides. *iso ΔBaraA* flies also fail to produce an immune-induced peak at ~5795 Da corresponding to IM22 (the C-terminal peptide of BaraA, see S1 Text). A diagram of the *ΔBaraA^SW1* mutation that replaces the N-terminal gene region with a DsRed construct is shown in the bottom right. **B)** The *BaraA* gene encodes a precursor protein that is cleaved into multiple mature peptides at RXRR furin cleavage sites. The sub-peptides IMs 5, 6, and 8 are additional minor cleavage products of IM12 and IM13. IM22 is additionally cleaved following its GIND motif (S2 Fig and S3A). **C)** There is a *BaraA* locus duplication event present in the Dmel_R6 reference genome. This duplication is not fixed in laboratory stocks and wild-type flies [25]. The *ΔBaraA* mutation was generated in a background with only one *BaraA* copy.

flies to LC-MS proteomic analysis following trypsin digestion and found that in addition to the known IMs of *BaraA* (IMs 5, 6, 8, 10, 12, 13, and 24), trypsin-digested fragments of the *BaraA* C-terminus peptide were also detectable in the haemolymph (S2 Fig). The range of detected fragments did not match the full length of the C-terminus exactly, as the first four residues were absent in our LC-MS data (a truncation not predicted to arise via trypsin cleavage). The *BaraA* C-terminus lacking these four residues has a calculated mass of 5974.5 Da, exactly matching the observed mass of the IM22 peak absent in *BaraA* mutant flies. Furthermore in other *Drosophila* species these four residues are absent, and instead the C-terminus directly follows an RXRR furin cleavage motif (S3A Fig). Therefore IM22 cleavage in other species, even by an alternate cleavage process, should result in the same matured IM22 domain as found in *D. melanogaster*. Taken together, we conclude that IM22 is the mature form of the BaraA protein C-terminus.

Thus, a single gene, *BaraA*, contributes to one third of the originally described *Drosophila* IMs. These peptides are encoded as a polypeptide precursor interspersed by furin cleavage sites (e.g. RXRR) (Fig 2B). We note that the IM10, IM12 and IM13 peptides are tandem repeats of related peptides, which we collectively refer to as "IM10-like" peptides (S3B Fig). The IM22 peptide also contains a similar motif as the IM10-like peptides (S3A and S3B Fig), suggesting a related biological activity. We name this gene "*Baramicin A*" (symbol: *BaraA*) for the Japanese idiom Bara Bara (バラバラ), meaning "to break apart;" a reference to the fragmenting structure of the *Baramicin* precursor protein and its many peptidic products.

## A *BaraA* duplication is present in some laboratory stocks

Over the course of our investigation, we realized that *IMPPP (CG18279)* was identical to its neighbour gene *CG33470* owing to a duplication event of the *BaraA* locus present in the *D. melanogaster* reference genome. The exact nature of this duplication is discussed in a separate article [25]. In brief, the duplication involves the entire *BaraA* gene including over 1kbp of 100% identical promoter sequence, and also the neighbouring sulfatase gene CG30059 and the 3' terminus of the *ATP8A* gene region (Fig 2C). We distinguish the two daughter genes as *BaraA1 (CG33470)* and *BaraA2 (CG18279)*. Available sequence data suggests the *BaraA1* and *BaraA2* transcripts are 100% identical. In a separate study, we analyzed the presence of the *BaraA* duplication using a PCR assay spanning the junction of the duplicated region (also see S3 Data). Interestingly, *BaraA* copy number is variable in common lab strains and wild flies, indicating this duplication event is not fixed in *D. melanogaster* [25].

## Over-expression of *BaraA* improves the resistance of immune deficient flies

*Imd*, *Toll* deficient flies are extremely susceptible to microbial infection as they fail to induce hundreds of immune genes, including antimicrobial peptides [11]. It has been shown that over-expression of even a single AMP can improve the resistance of *Imd*, *Toll* deficient flies [26]. As such, immune gene over-expression in *Imd*, *Toll* immune-compromised flies provides a direct assay to test the ability of a gene to contribute to defence independent of other immune effectors. We applied this strategy to *Baramicin A* by generating flies that constitutively express *BaraA* using the ubiquitous *Actin5C-Gal4* driver *(Act-Gal4)* in an immune-deficient *Rel^E20^*, *spz^rm7^* double mutant background (S4A Fig). In these experiments, we pooled results from both males and females due to the very low availability of homozygous *Rel*, *spz* adults when combined with *Act-Gal4*. Overall, similar trends were seen in both sexes, and separate male and female survival curves are shown in S4 Fig.

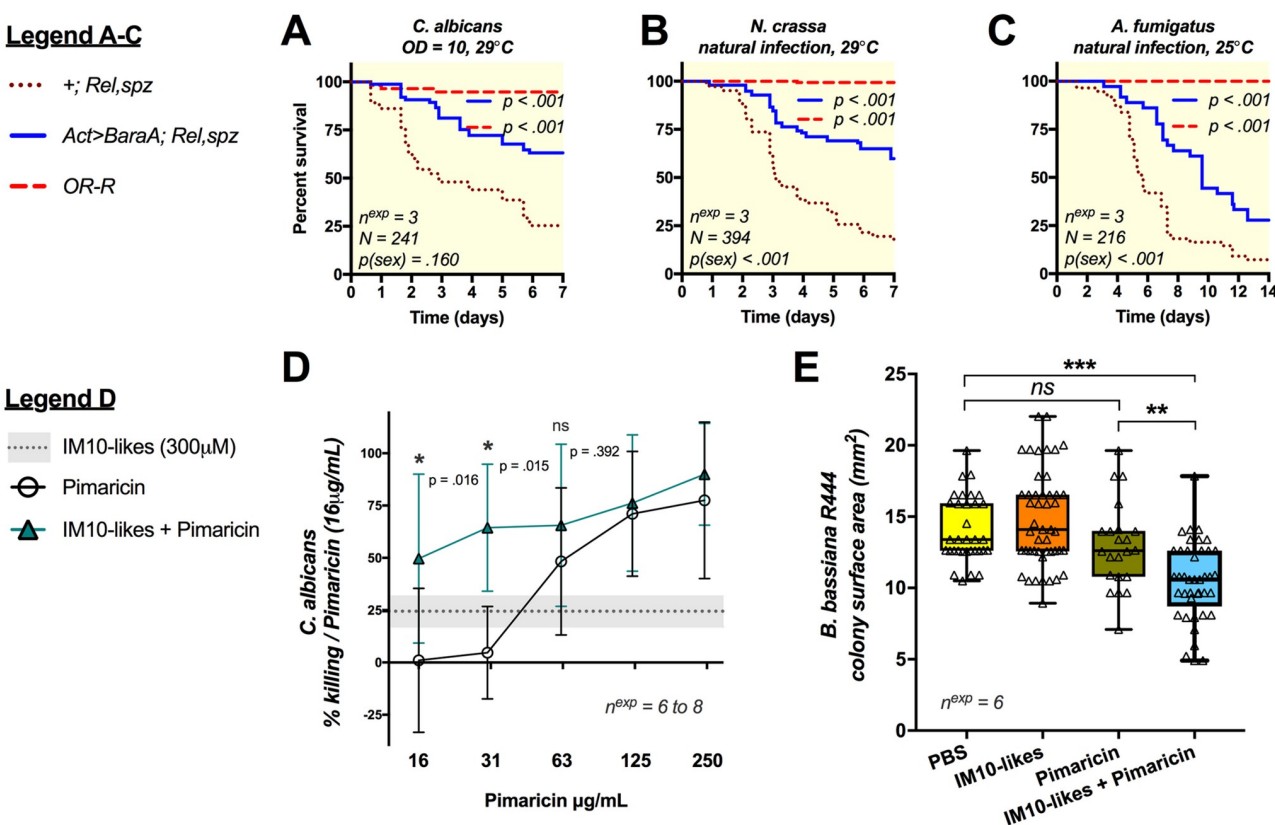

**Fig 3. Overexpression of *BaraA* partially rescues the susceptibility of *Rel*, *spz* flies against fungi and BaraA IM10-like peptides inhibit fungal growth in vitro. A-C)** Overexpression of *BaraA* (*Act>BaraA*) rescues the susceptibility of *Rel*, *spz* flies upon systemic infection with *C. albicans* (**A**), or natural infection with either *N. crassa* or *A. fumigatus* (**B-C**). Survivals represent pooled results from males and females (see S4 Fig for sex-specific survival curves). **D)** A 300μM cocktail of the three IM10-like peptides improves the killing activity of the antifungal Pimaricin against *C. albicans* yeast. Error bars and the shaded area (IM10-likes alone) represent ±1 standard deviation from the mean. Killing activity (%) was compared against no-peptide controls, then normalized to the activity of Pimaricin alone. **E)** The IM10-like peptide cocktail also synergizes with Pimaricin (250μg/mL) to inhibit mycelial growth of *B. bassiana strain R444*. The diameters of individual colonies of *B. bassiana* were assessed after four days of growth at 25˚C after peptide treatment, and surface area calculated as πr².

Ubiquitous *BaraA* expression marginally improved the survival of *Rel*, *spz* flies upon infection with *M luteus* bacteria, however there was no effect upon infection with *E. coli* (S4B and S4C Fig). On the other hand, ubiquitous expression of *BaraA* provided a more pronounced protective effect against infection by a variety of fungal pathogens. This was true upon pricking with *C. albicans* (Fig 3A), or upon natural infections using *Aspergillus fumigatus* or *Neurospora crassa* filamentous fungi (Fig 3B and 3C). This over-expression study reveals that *BaraA* alone can partially rescue the susceptibility of *Imd*, *Toll* deficient flies to infection, and points to a more prominent role for *BaraA* in antifungal defence.

## IM10-like peptides display antifungal activity in vitro

*The Baramicin A* gene encodes a polypeptide precursor that ultimately produces multiple mature peptides. However the most prominent *BaraA* products are the 23-residue IM10, 12, and 13 peptides (collectively the "IM10-like" peptides); indeed three IM10-like peptides are produced for every one IM24 peptide (Fig 2B), and IM22 also bears an IM10-like motif (S3 Fig). This prompted us to explore the in vitro activity of the BaraA IM10-like peptides as potential AMPs.

We synthesized IM10, IM12, and IM13 and performed in vitro antimicrobial assays with these three IM10-like peptides using a 1:1:1 cocktail with a final concentration of 300μM (100 μM each of IM10, IM12, and IM13). We monitored the microbicidal activity of this peptide cocktail using a protocol adapted from Wiegand et al. [27]. We did not detect any killing activity of our IM10-like peptide cocktail alone against *Pectobacterium carotovora Ecc15* (hereafter *"Ecc15"*), *Enterococcus faecalis*, or *C. albicans*. Previous studies have shown that the microbicidal activities of Abaecin-like peptides, which target the bacterial DNA chaperone *DnaK*, increase exponentially in combination with a membrane disrupting agent [28–30]. Inspired by this approach, we next assayed combinations of the IM10-like cocktail with membrane-disrupting antibiotics relevant to tested microbes that should facilitate peptide entry into the cell. We again found no activity of IM10-like peptides against *Ecc15* or *E. faecalis* when co-incubated with a sub-lethal dose of Cecropin or Ampicillin respectively, indicating IM10-like peptides likely do not affect *Ecc15* or *E. faecalis* either alone or in combination with membrane-disrupting antibiotics. However, we observed a synergistic interaction between IM10-like peptides and the commercial antifungal Pimaricin against *C. albicans* (Fig 3D). Co-incubation of the IM10-like cocktail with Pimaricin significantly improved the killing activity of Pimaricin at 16 and 32μg/mL relative to either treatment alone. While not statistically significant, the combination of IM10-like cocktail and Pimaricin also outperformed either the IM10-like cocktail alone or Pimaricin alone across the entire range of Pimaricin concentrations tested.

We next co-incubated dilute preparations of *B. bassiana* strain R444 spores under the same conditions as used previously with *C. albicans*, plated 2μL droplets, and assessed the diameters and corresponding surface area of colonies derived from individual spores after 4 days of growth at 25˚C to assess growth rate. We found that neither the IM10-like cocktail nor Pimaricin alone significantly affected surface area relative to a PBS buffer control (Tukey's HSD: p = 0.656 and 0.466 respectively). However in combination, the IM10-like cocktail plus Pimaricin led to significantly reduced colony size compared to either treatment alone, corresponding to a 19–29% reduction in surface area relative to controls (Fig 3E, Tukey's HSD: p < .01 in all cases). This indicates that incubation with IM10-like peptides and Pimaricin synergistically inhibits *B. bassiana* mycelial growth, revealing an otherwise cryptic antifungal effect of the BaraA IM10-like peptides in vitro.

Overall, we found that IM10-like peptides alone do not kill *C. albicans* yeast or impair *B. bassiana* mycelial growth in vitro. However, IM10-like peptides seem to synergize with the antifungal Pimaricin to inhibit growth of both of these fungi.

## *BaraA* deficient flies broadly resist like wild-type upon bacterial infection

To further characterize *BaraA* function, we generated a null mutation of *BaraA* by replacing the 'entire' *BaraA* locus with a dsRed cassette using CRISPR mediated homology-directed repair with fly stocks that contain only one *BaraA* gene copy (BDSC #2057 and BL51323) (Fig 2A). After isolation, this mutation ($BaraA^{SW1}$) was then backcrossed once to a lab strain of $w^{1118}$ (used in [13–15]) to remove a second site mutation (see Materials and methods). The resulting $w^{1118}$; $BaraA^{SW1}$ flies are hereon referred to as "w; ΔBaraA." As a consequence of this backcrossing event, *w; ΔBaraA* flies are a mixed genetic background, which we arbitrarily compare to *OR-R* as representative wild-type flies. Finally, the $BaraA^{SW1}$ mutation was isogenized by seven rounds of backcrossing into the $w^{1118}$ *DrosDel* isogenic genetic background (*iso $w^{1118}$*) [31] as described in Ferreira et al. and are hereon referred to as "*iso ΔBaraA*" [32]. Relevant to this study, both our *OR-R* and *DrosDel iso $w^{1118}$* wild-type lines contain the *BaraA* duplication and thus have both *BaraA1 and A2* genes, while *w; ΔBaraA* and *iso ΔBaraA* flies

lack *BaraA* entirely. In the following experiments, we compare the immune response of both *w; ΔBaraA* and *iso ΔBaraA* to wild-type flies, and focused on phenotypes that were consistent in both genetic backgrounds.

We validated these mutant lines by PCR, qPCR and MALDI-TOF peptidomics (Fig 2A and S3 Data). *BaraA*-deficient flies were viable with no morphological defects. Furthermore, *ΔBaraA* flies have wild-type Toll and Imd signalling responses following infection, indicating that *BaraA* is not required for the activation of these signaling cascades (S5A–S5C Fig). *BaraA* mutant flies also survive clean injury like wild-type (S5D Fig), and have comparable lifespan to wild-type flies (S5E Fig). We next challenged *BaraA* mutant flies using our two genetic backgrounds with a variety of pathogens. We included susceptible Imd deficient *Rel*[E20] flies, Toll deficient *spz*[rm7] flies and *Bomanin* deficient *Bom*[Δ55C] flies as comparative controls. We observed that *BaraA* null flies have comparable resistance as wild-type to infection with the Gram-negative bacteria *Ecc15* and *Providencia burhodogranariea* (S6A and S6B Fig), or with the Gram-positive bacterium *Bacillus subtilis* (S6C Fig). In contrast, we saw a mild increase in the susceptibility of *w; ΔBaraA* flies to infection by the Gram-positive bacterium *E. faecalis* (HR = +0.73, p = .014). We also saw an early mortality phenotype in *iso ΔBaraA* flies (at 3.5 days, p < .001), although this was not ultimately statistically significant (S7A Fig; p = .173). This trend of a mild susceptibility was broadly consistent in deficiency crosses and flies ubiquitously expressing *BaraA* RNAi (S7B and S7C Fig), though none of these sets of survival experiments individually reached statistical significance. Overall, the susceptibility of *BaraA* mutants to *E. faecalis* is mild, but appears consistent using a variety of genetic approaches.

## *BaraA* mutant flies are highly susceptible to *Beauveria* fungal infection

Entomopathogenic fungi such as *Metarhizium* and *Beauveria* represent an important class of insect pathogens [6]. They have the ability to directly invade the body cavity by digesting and crossing through the insect cuticle. The Toll pathway is critical to survive fungal pathogens as it is directly responsible for the expression of *Bomanin*, *Daisho*, *Drosomycin* and *Metchnikowin* antifungal effectors [13,15,16,33,34]. The fact that i) *BaraA* is Toll-regulated, ii) BaraA IM10-like peptides display antifungal activity in vitro, and iii) *BaraA* overexpression improves the resistance of *Imd*, *Toll* deficient flies against fungi all point to a role for *BaraA* against fungal pathogens.

We infected *BaraA* mutant and wild-type flies using a septic injury model of *Metarhizium rileyi* strain PHP1705 (Andermatt Biocontrol). *spz*[rm7] and *Bom*[Δ55C] mutant flies were highly susceptible to *M. rileyi* septic injury. Likewise, both *w; ΔBaraA* and *iso ΔBaraA* mutants showed a significant susceptibility to *M. rileyi* septic injury (Fig 4A, HR ≥ 1.0 and p < .05 in both cases). We next rolled flies in sporulating *B. bassiana strain 802* petri dishes. Strikingly, both *w; ΔBaraA* and *iso ΔBaraA* flies displayed a pronounced susceptibility to natural infection with *B. bassiana* (HR = +2.10 or +0.96 respectively, p < .001 for both) (S8A Fig). An increased susceptibility to fungi was also observed using flies carrying the *BaraA* mutation over a deficiency (S8B Fig) or that ubiquitously express *BaraA* RNAi (S8C Fig). Moreover, constitutive *BaraA* expression (*Act-Gal4>UAS-BaraA)* in an otherwise wild-type background improves survival to *B. bassiana 802* relative to *Act-Gal4>OR-R* controls (HR = -0.52, p = .010) (S8D Fig). We next used a preparation of commercial *B. bassiana R444* spores (Andermatt Biocontrol) to perform controlled systemic infections by septic injury with a needle dipped in spore solution. In these experiments we monitored both survival and fungal load using qPCR primers specific to the *B. bassiana* 18S rRNA gene [35]. As seen with natural infection, *BaraA* mutants were highly susceptible to *Beauveria* systemic infection (Fig 4B), and suffered increased fungal load by 48 hours after infection (Fig 4C). We also compared the effect of

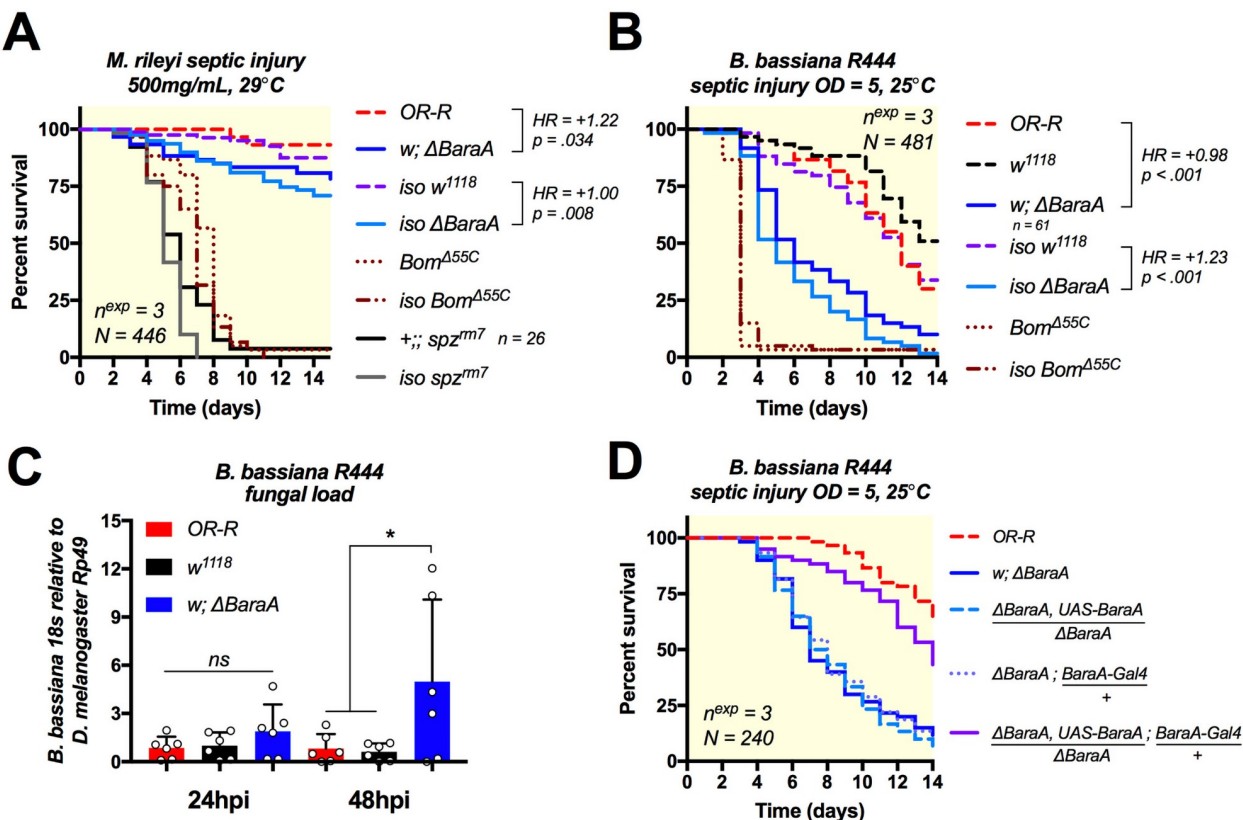

**Fig 4. *ΔBaraA* flies are susceptible to fungal infection. A)** *BaraA* mutants in two genetic backgrounds (here called *w* or *iso*) display a significant susceptibility to septic injury with *M. rileyi*. **B-C)** Increased susceptibility of *ΔBaraA* flies upon septic injury with *B. bassiana* R444 (**B**) correlates with increased fungal load 48hpi (**C**). **D)** Heterologous expression of *BaraA* via combination of the *BaraA-Gal4* and *UAS-BaraA* constructs rescues the susceptibility of *BaraA* mutant females to *B. bassiana* infection.

*BaraA* in defence against *B. bassiana* to the effect of deleting two classical antifungal peptide genes of *Drosophila*: *Metchnikowin (Mtk)* and *Drosomycin (Drs)*. Use of infection models with very different virulence (septic injury vs. natural infection) suggests that *BaraA* contributes far more strongly to defence against *B. bassiana* compared to the combined effect of *Mtk* and *Drs* (S8E Fig), while *Mtk* and *Drs* did not greatly affect resistance relative to wild-type (HR = +0.15, p >.10).

Finally, we combined the *ΔBaraA* mutation with both a *UAS-BaraA* construct on the 2nd chromosome or our *BaraA-Gal4* driver on the 3rd chromosome to rescue the susceptibility of *BaraA* deficient flies. Supplementing *ΔBaraA* flies with *BaraA* expressed via the *BaraA-Gal4>UAS-BaraA* method restores resistance almost to wild-type levels (Fig 4D). Collectively, our survival analyses point to a role for *BaraA* in defence against entomopathogenic fungi, including *M. rileyi* and especially *B. bassiana*. Consistent with a direct effect of *BaraA* on fungi, *BaraA* mutant susceptibility is correlated with increased proliferation of *B. bassiana*, and heterologous expression of *BaraA* via the Gal4/UAS system rescues the susceptibility of mutants, confirming that mutant susceptibility is caused by the loss of *BaraA*.

## *BaraA* contributes to antifungal defence independent of *Bomanins*

Use of compound mutants carrying multiple mutations in effector genes has shown that some of them additively contribute to host resistance to infection [16]. Compound deletions of

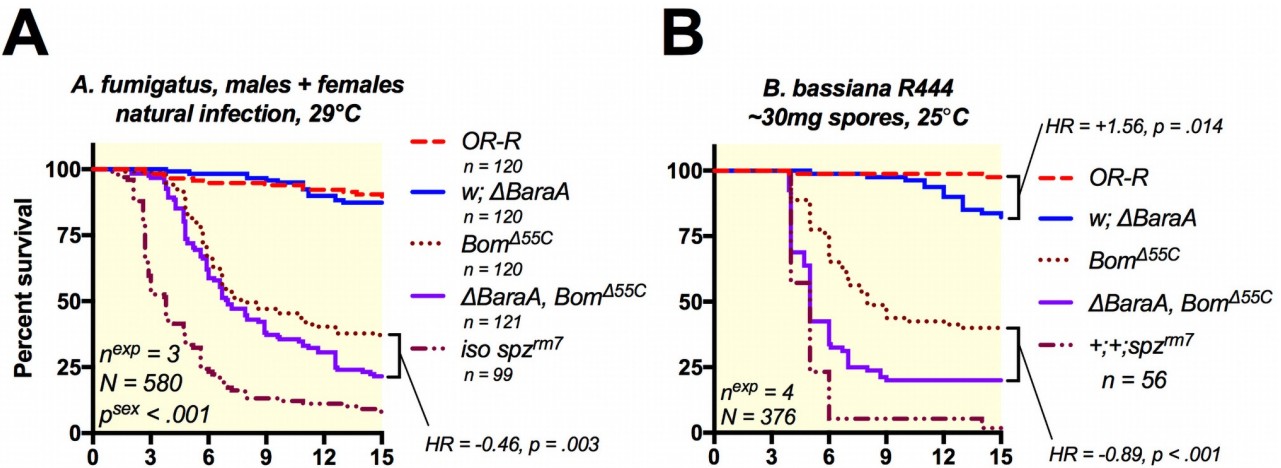

**Fig 5. *BaraA* contributes to antifungal defence independent of other effectors. A)** *ΔBaraA, Bom^{Δ55C}* double mutant flies were more susceptible than either mutation alone to natural infection with *A. fumigatus* (see S6D and S6E Fig for sex-specific survival curves). **B)** *ΔBaraA, Bom^{Δ55C}* double mutant flies were similarly more susceptible than individual mutants when given a mild (30mg of spores) *Beauveria* natural infection using *B. bassiana R444*.

immune genes can also reveal contributions of immune effectors that are not detectable via single mutant analysis [16,36,37]. Recent studies have indicated that *Bomanins* play a major role in defence against fungi [13,14], though their mechanism of action is unknown. It is possible that *Bomanin* activity relies on the presence of *BaraA*, or vice versa. This prompted us to investigate the interaction of *Bomanins* and *BaraA* in defence against fungi. To do this, we recombined the *Bom^{Δ55C}* mutation (that removes a cluster of 10 *Bomanin* genes) with *ΔBaraA*. Furthermore, we used low-virulence models of infection that allowed some *Bomanin* mutant flies to survive, so as to ensure additional mutation of *BaraA* had an opportunity to affect survival if relevant. While natural infection with *Aspergillus fumigatus* did not induce significant mortality in *BaraA* single mutants (S6D and S6E Fig), we observed that combining *ΔBaraA* and *Bom^{Δ55C}* mutations increases fly susceptibility to this pathogen relative to *Bom^{Δ55C}* alone (HR = -0.46, p = .003; Fig 5A). We next exposed these *ΔBaraA, Bom^{Δ55C}*, double mutant flies to a low dose natural infection with 30mg of commercial spores of *B. bassiana R444* as this dose allows some *Bomanin* mutant flies to survive. This is equivalent to approximately 60 million spores added to a vial containing 20 flies, many of which are removed afterwards upon fly grooming. When using this infection method, we found that *BaraA* mutation markedly increases the susceptibility of *Bom^{Δ55C}* mutant flies (HR = -0.89, p < .001), approaching *spz^{rm7}* susceptibility (Fig 5B).

If BaraA and Bom peptides relied on each other for activity, we would expect no increased susceptibility of double mutants. However *BaraA, Bom* double mutation results in increased susceptibility relative to *Bom* mutation alone. We conclude *BaraA* acts independently of *Bomanins*, agreeing with the ability of heterologous overexpression of *BaraA* to rescue Toll, Imd double mutant flies that are similarly deficient in *Bomanin* production (Fig 3A–3C). Alongside a more prominent activity of *BaraA* in defence against *B. bassiana* compared to *Drs* and *Mtk* (S8D Fig), these results suggest *BaraA* improves survival against fungi independent of other effectors of the systemic immune response also using effector mutant analysis, consistent with a direct effect on invading fungi.

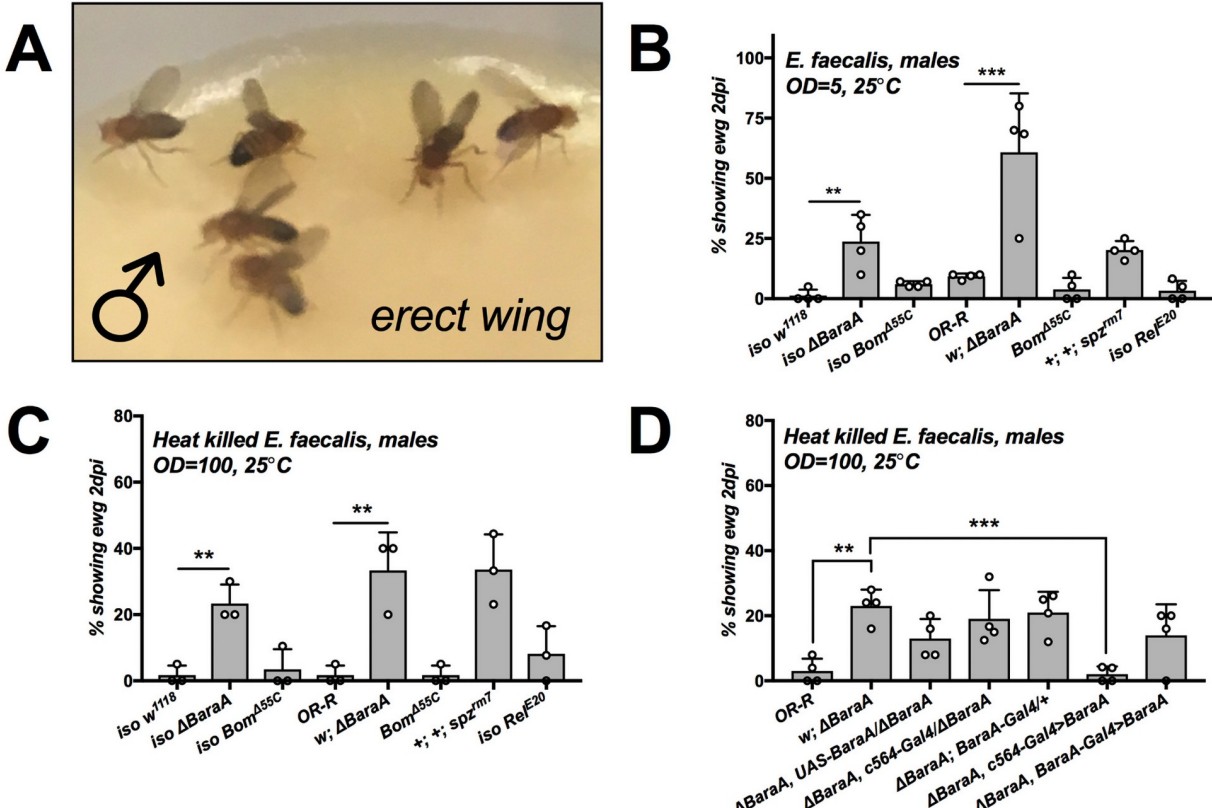

**Fig 6. *ΔBaraA* males display an erect wing phenotype upon infection. A)** *ΔBaraA* males displaying erect wing six days after *A. fumigatus* natural infection. **B-C)** *spz^rm7* and *ΔBaraA* males, but not *Bom^Δ55C* or *Rel^E20* flies display the erect wing phenotype upon septic injury with live (**B**) or heat-killed *E. faecalis* (C). **D)** The presentation of erect wing in *ΔBaraA* flies is rescued by c564-Gal4 ubiquitous expression of *BaraA*. Barplots show the percentage of flies displaying erect wing following treatment, with individual data points reflecting replicate experiments. Asterisks indicate one-way ANOVA significance relative to reference *w; ΔBaraA* flies (**, and *** = p < .01, and.001 respectively). Erect wing frequency after additional challenges are shown in S9 Fig and S1 Table.

## *ΔBaraA* males display an erect wing phenotype upon infection

While performing natural infections with *A. fumigatus*, we observed a high prevalence of *BaraA* mutant flies with upright wings (Fig 6A and S9A Fig), a phenotype similar to the effect of disrupting the gene encoding the "*erect wing*" (*ewg*) transcription factor [38]. Curiously, this erect wing phenotype was most specifically observed in males. Upon further observation, erect wing was observed not only upon *A. fumigatus* infection, but also upon infections with all Gram-positive bacteria and fungi tested, and less so upon clean injury or using Gram-negative bacteria (S1 Table and S9B and S9C Fig). We eventually pursued this striking phenotype further using an *E. faecalis* septic injury model. A greater prevalence of erect wing flies was observed upon infection with live *E. faecalis* (Fig 6B). Strikingly, even injury with heat-killed *E. faecalis* is sufficient to induce erect wing (Fig 6C), collectively indicating that this phenotype is observed in *BaraA* mutants upon Toll pathway stimulation, but does not require a live infection.

Such a phenotype in infected males has never been reported, but is reminiscent of the wing extension behaviour of flies infected by the brain-controlling "zombie" fungus *Enthomopthera muscae* [39]. Intrigued by this phenotype, we further explored its prevalence in other genetic backgrounds. We next confirmed that this phenotype was also observed in other

*BaraA*-deficient backgrounds such as *Df(BaraA)/ΔBaraA*; however the penetrance was variable from one background to another (S1 Table). Erect wing was also observed in *ΔBaraA/+* heterozygous flies (*Df(BaraA)/+* or *ΔBaraA/+*), indicating that the lack of *BaraA* on one chromosome was sufficient to cause the phenotype (S9D Fig), independent of overall susceptibility to *E. faecalis* (S7B Fig). Moreover, *spz*[rm7] flies that lack functional Toll signalling phenocopy *ΔBaraA* flies and display erect wing, but other immune-deficient genotypes such as mutants for the Toll-regulated *Bomanin* effectors (*Bom*[Δ55C]), or *Rel*[E20] mutants that lack Imd signalling, did not readily display erect wing (Fig 6B and 6C and S1 Table). Thus the erect wing phenotype is not linked to susceptibility to infection, but rather to loss of *BaraA* alongside stimuli triggering the Toll immune pathway. This phenotype suggests an additional effect of *BaraA* on tissues related to the wing muscle or in the nervous system.

The expression profile of *BaraA* is complex and poorly defined in existing transcriptomic datasets, likely owing to the gene duplication of *BaraA1* and *BaraA2* complicating read mapping [22,25]. As *BaraA* is expressed in various tissues including the head/eye, crop, and fat body (Fig 1 and [23]), it is unclear if *BaraA* absence in the brain, neuromusculature, or non-neuronal tissues (such as the fat body) could underlie the predisposal to erect wing. To this end, we used stocks containing both the *ΔBaraA* mutation and either a *UAS-BaraA* construct, *c564-Gal4* constitutive fat body driver, or *BaraA-Gal4* driver, and performed genetic crosses to attempt to rescue the presentation of erect wing upon septic injury with heat-killed *E. faecalis* using the Gal4/UAS system. Surprisingly, constitutive *BaraA* expression in the fat body by *c564-Gal4* rescued erect wing presentation to effectively zero levels (Fig 6D). On the other hand, *ΔBaraA, BaraA-Gal4>UAS-BaraA* flies displayed erect wing (exact genotype as in Fig 4E), similar to *Df(BaraA)/+* and *ΔBaraA/+* flies (S9D Fig). Indeed, qPCR of *BaraA* expression after infection shows that *BaraA* levels are lower than wild-type in both *ΔBaraA, BaraA-Gal4>UAS-BaraA* (S9E Fig) and *ΔBaraA/+* transheterozygotes (S10 Fig).

Cumulatively, these experiments confirm that loss of *BaraA* results in the erect wing phenotype upon immune stimulus given *BaraA* deficiency, either by mutation or by loss of Toll signalling. This phenotype occurs independent of active infection, and is specifically tied to *BaraA* downstream of Toll signalling. A full transcriptional output of *BaraA* appears to be required to prevent erect wing after infection, as flies with less than wild-type *BaraA* expression are predisposed to displaying erect wing. However priming the hemolymph with BaraA peptides via constitutive expression in the fat body is sufficient to rescue the erect wing phenotype. Importantly, this rescue by fat body driven expression indicates that systemically secreted BaraA peptides mediate this phenotype, and not *BaraA* expression in e.g. neuronal tissue. Taken together, a wild-type induction of *BaraA* is required to prevent erect wing presentation following Toll activation, which can be ameliorated by priming the hemolymph with constitutive *BaraA* expression.

## Discussion

Seven *Drosophila* AMP families were identified in the 1980s-1990s either by homology with AMPs characterized in other insects or owing to their abundant production and microbicidal activities in vitro [40]. In the 2000s, genome annotations revealed the existence of many additional paralogous genes from the seven well-defined families of AMPs [41,42]. At that time, microarray and MALDI-TOF analyses also revealed the existence of many more small immune-induced peptides, which may function as AMPs [8,24]. Genetic analyses using loss of function mutations have recently shown that some of these peptides do play an important role in host defence, however key points surrounding their direct microbicidal activities remain unclear. In 2015, *Bomanins* were shown to be critical to host defence using genetic approaches,

but to date no activity in vitro has been found [13,14]. The overt susceptibility of Bomanin mutants to most Gram-positive bacteria and fungi also suggests a generalist role in supporting the effectors of Toll, rather than a direct effect on microbes. In addition, two candidate AMPs, Listericin [43] and GNBP-like3 [44], have been shown to inhibit microbial growth upon heterologous expression using S2 cell lines or bacteria respectively. Most recently, Daisho peptides were shown to bind to fungal hyphae ex vivo, and are required for resisting *Fusarium* fungal infection in vivo [15]. However the mechanism and direct microbicidal activity of these various peptides at physiological concentrations has not yet been assessed.

In this study, we provide evidence from four separate experimental approaches that support adding *BaraA* products to the list of bona-fide antifungal peptides. First, the *BaraA* gene is strongly induced in the fat body upon infection resulting in abundant peptide production. *BaraA* is also tightly regulated by the Toll pathway, which orchestrates the antifungal response. Second, loss of function study shows that *BaraA* contributes to resistance against fungi. *BaraA* mutation increases susceptibility to *M. rileyi* and *B. bassiana*, and *BaraA* deficient flies suffer increased *B. bassiana* proliferation. Third, the antifungal activity of *BaraA* is independent of other key effectors. Over-expression of *BaraA* in the absence of Toll/Imd inducible peptides increased the resistance of compound *Rel*, *spz* deficient flies to various fungi including *C. albicans*, *A. fumigatus*, and *N. crassa*, and rescues the *ΔBaraA* mutant susceptibility to *B. bassiana*. Additionally, compound gene deletion of both *BaraA and Bomanins* causes greater susceptibility than *Bomanin* mutation alone after *B. bassiana* natural infection. Fourth, and lastly, a cocktail of the *BaraA* IM10-like peptides possesses antifungal activity against *C. albicans* and *B. bassiana* in vitro when co-incubated with the membrane disrupting antifungal Pimaricin.

While it is difficult to estimate the concentration of BaraA peptides in the haemolymph of infected flies, it is expected based on MALDI-TOF peak intensities that the IM10-like peptides should reach concentrations similar to other AMPs (up to 100μM) [10,21]; our in vitro assays used a peptide cocktail at the upper limit of this range. AMPs are often—but not exclusively–positively charged. This positive charge is thought to recruit these molecules to negatively charged membranes of microbes [10]. That said, the net charges at pH = 7 of the IM10-like peptides are: IM10 +1.1, IM12 +0.1, and IM13–0.9. Given this range of net charge, IM10-like peptides are not overtly cationic. However some AMPs are antimicrobial without being positively charged, exemplified by human Dermicidin [45] and anionic peptides of Lepidoptera that synergize with membrane-disrupting agents [46]. More extensive in vitro experiments with additional fungi and alternate membrane-disrupting antifungals (such as other insect or *Drosophila* antifungal peptides) should confirm the range of BaraA peptide activities. Furthermore, the potential activities of IM22 and IM24 should be addressed, which were not included in the present study. Future studies would benefit from testing different in vitro approaches, which might better mimic physiological conditions that could be relevant for BaraA peptide activity.

Our study also reveals that the *Baramicin A* gene alone produces at least 1/3 of the initially reported IMs. In addition to the IM10-like peptides and IM24 that were previously assigned to *BaraA* [24], we show IM22 is encoded by the C terminus of *BaraA*, and is conserved in other *Drosophila* species. The production of multiple IMs encoded as tandem repeats between furin cleavage sites is built-in to the BaraA protein design akin to a "protein operon." Such tandem repeat organization is rare, but not totally unique among AMPs. This structure was first described in the bumblebee AMP Apidaecin [47], and has since also been found in Drosocin of *Drosophila neotestacea* [48]. In *D. melanogaster*, several AMPs are furin-processed including Attacin C and its pro-peptide MPAC, wherein both parts synergize in killing bacteria [28]. Therefore, furin cleavage in Attacin C enables the precise co-expression of distinct peptides with synergistic activity. It is interesting to note that IM10-like peptides did not show

antifungal activity in the absence of membrane disruption by Pimaricin. An attractive hypothesis is that longer peptides encoded by BaraA such as IM22 and IM24 could contribute to the antifungal activity of *BaraA* by membrane permeabilization, allowing the internalization of IM10-like peptides. However rigorous experimentation is needed to determine the IM10-like mechanism of action. Indeed, the BaraA IM24 peptide is a short Glycine-rich peptide (96 AA) that is positively-charged (charge +2.4 at pH = 7). These traits are shared by amphipathic membrane-disrupting AMPs such as Attacins [10], however the precise role of the Baramicin IM24 domain is likely complex given the repeated evolution of neural-specific *Baramicins* that preferentially retain the IM24 domain [25].

An unexpected observation of our study is the display of an erect wing phenotype by *BaraA* deficient males upon infection. Our study suggests that this phenotype relies on the activation of the Toll pathway in the absence of *BaraA*. Erect wing is also induced by heat-killed bacteria, and is not observed in *Bomanin* or *Relish* mutants, indicating that the erect wing phenotype is not a generic consequence of susceptibility to infection. The *erect wing* gene, whose inactivation causes a similar phenotype, is a transcription factor that regulates synaptic growth in developing neuromuscular junctions [38]. This raises the intriguing hypothesis that immune processes downstream of the Toll ligand Spaetzle somehow affect wing neuromuscular junctions, and that *BaraA* modulates this activity. Another puzzling observation is the sexual dimorphism exhibited for this response. Male courtship and aggression displays involve similar wing extension behaviours. Koganezawa et al. [49] showed that males deficient for *Gustatory receptor 32a (Gr32a)* failed to unilaterally extend wings during courtship display. *Gr32a*-expressing cells extend into the subesophageal ganglion where they contact mAL, a male-specific set of interneurons involved in unilateral wing display [49]. One possible explanation for the male specific effects of *BaraA* could be that *BaraA* mediates this effect through interactions with such male-specific neurons. Recent studies have highlighted how NF-κB signalling in the brain is activated by bacterial peptidoglycan [50], and that immune effectors expressed either by fat body surrounding the brain or from within brain tissue itself affect memory formation [44]. Moreover, an AMP of nematodes regulates aging-dependent neurodegeneration through binding to its G-protein coupled receptor, and this pathway is sufficient to trigger motor neuron degeneration following infection [51]. The ability of fat body-derived *BaraA* to rescue the erect wing phenotype suggests a similar interplay of the immune response with neuromuscular processes. Future studies characterizing the role of *BaraA* in the erect wing phenotype should provide insight on interactions between systemic immunity and host physiology more generally.

Here we describe a complex immune effector gene that produces multiple peptide products. *BaraA* encodes many of the most abundant immune effectors induced downstream of the Toll signalling pathway. We show that *BaraA* has a pronounced effect on survival after *Beauveria* fungal infection. Moreover, this gene regulates an erect wing behavioural response upon infection. How each peptide contributes to the immune response and/or erect wing behaviour will be informative in understanding the range of effects immune effectors can have on host physiology. This work and others also clarifies how the cocktail of immune effectors produced upon infection acts specifically during innate host defence reactions.

## Materials and methods

### Fly genetics and sequence comparisons

Sequence files were collected from FlyBase [52] and recently-generated sequence data [48,53] and comparisons were made using Geneious R10. Putative NF-κB binding sites were annotated using the Relish motif "GGRDNNHHBS" described in Copley et al. [19] and a manually

curated amalgam motif of "GGGHHNNDVH" derived from common Dif binding sites described previously [18,20]. Gene expression analyses were performed using primers described in S3 Data, and further microarray validation for *BaraA* expression comes from De Gregorio et al. [11].

The *UAS-BaraA* and *BaraA-Gal4* constructs were generated using the TOPO pENTR entry vector and cloned into the pTW or pBPGUw Gateway vector systems respectively. The *BaraA-Gal4* promoter contains 1675bp upstream of *BaraA1* (but also *BaraA2*, sequence in S1 Text). The *BaraA-Gal4* construct was inserted into the VK33 attP docking site (BDSC line #24871). The *BaraA^{SW1}* (*ΔBaraA)* mutation was generated using CRISPR with two gRNAs and an HDR vector by cloning 5' and 3' region-homologous arms into the pHD-dsRed vector, and consequently *ΔBaraA* flies express dsRed in their eyes, ocelli, and abdomen. *ΔBaraA* was generated using the Bloomington stocks BL2057 and BL51323 as these backgrounds contain only one copy of the *BaraA* locus. The induction of the immune response in these flies was validated by qPCR and MALDI-TOF proteomics, wherein we discovered an aberrant *Dso2* locus in these preliminary *BaraA^{SW1}* flies. We thus backcrossed the *BaraA^{SW1}* mutation once with a standard *w^{1118}* background (used in [13–15]) and screened for wild-type *Dso2* before use in any survival experiments. As a consequence, *w; ΔBaraA* flies are considered an arbitrary genetic background with no appropriate wild-type control. We typically used *Oregon-R* (*OR-R*) flies as a representative wild-type that displays similar resistance to bacterial infections (S6 Fig). Of note, *ΔBaraA* was also isogenized into the *DrosDel w^{1118}* isogenic background for seven generations before use in isogenic fly experiments as described in Ferreira et al. [32]. We value the use of both genetic backgrounds to ensure that interpretation of mutant analysis is not biased by genetic background.

A full description of fly stocks used for crosses and in experiments is provided in S4 Data.

## Microbe culturing conditions

Bacteria and *C. albicans* yeast were grown to mid-log phase shaking at 200rpm in their respective growth media (Luria Bertani, Brain Heart Infusion, or Yeast extract-Peptone-Glycerol) and temperature conditions, and then pelleted by centrifugation to concentrate microbes. Resulting cultures were diluted to the desired optical density at 600nm (OD) for survival experiments, which is indicated in each figure. The following microbes were grown at 37˚C: *Escherichia coli strain 1106* (LB), *Enterococcus faecalis* (BHI), and *Candida albicans* (YPG). The following microbes were grown at 29˚C: *Erwinia carotovora carotovora (Ecc15)* (LB) and *Micrococcus luteus* (LB). For filamentous fungi and molds, *Aspergillus fumigatus* was grown at 37˚C, and *Neurospora crassa* and *Beauveria bassiana strain 802* were grown at room temperature on Malt Agar in the dark until sporulation. *Metarhizium rileyi strain PHP1705* and *Beauveria bassiana strain R444* commercial spores were produced by Andermatt Biocontrol, products: Nomu-PROTEC and BB-PROTEC respectively. A summary of microbe strains is provided in S4 Data.

## Survival experiments

Survival experiments were performed as previously described [16], with 20 flies per vial with 2–3 replicate experiments. 3–5 day old males were used in experiments unless otherwise specified. As *Rel*, *spz* double mutant flies and wild-type backgrounds differ drastically in their immune competence, we selected pathogens, infection routes, and temperatures to provide infection models that could best reveal phenotypes in these disparate genetic backgrounds. For fungi natural infections, flies were flipped at the end of the first day to remove excess fungal spores from the vials. Otherwise, flies were flipped thrice weekly. Statistical analyses were

performed using a Cox proportional hazards (CoxPH) model in R 3.6.3. We report the hazard ratio (HR) alongside p-values as a proxy for effect size in survival experiments. Throughout our analyses, we required $p < .05$ as evidence to report an effect as significant, but note interactions with $|HR|$ near or above 0.5 as potentially important provided p-value approached .05, and tamp down importance of interactions that were significant, but have relatively minor effect size ($|HR|$ less than 0.5) in our discussion of the data.

## Erect wing scoring

The erect wing phenotype was scored as the number of flies with splayed wings throughout a distinct majority of the period of observation (30s); if unclear, the vial was monitored an additional 30s. Here we define splayed wings as wings not at rest over the back, but did not require wings to be fully upright; on occasion wings were held splayed outward at ~45˚ relative to the dorsal view, and often slightly elevated relative to the resting state akin to male aggressive displays. Sometimes only one wing was extended, which occurred in both thoracic pricking and fungi natural infections; these flies were counted as having erect wing. In natural infections, the typical course of erect wing display developed in two fashions at early time points, either: i) flies beginning with wings slightly splayed but not fully upright, or ii) flies constantly flitting their wings outward and returning them to rest briefly, only to flit them outward again for extended periods of time. Shortly after infection, some flies were also observed wandering around with wings beating at a furious pace, which was not counted as erect wing. However at later time points erect wing flies settled more permanently on upright splayed wings. Erect wing measurements were taken daily following infection, and erect wing flies over total flies was converted to a percent. Data points in Fig 6B–6D represent % with erect wing in individual replicate experiments with 20–25 flies per vial. Flies stuck in the vial, or where the wings had become sticky or mangled were not included in totals. S1 Table reports mean percentages across replicate experiments for all pathogens and genotypes where erect wing was monitored. Days post-infection reported in S1 Table were selected as the final day prior to major incidents of mortality. For *E. faecalis* live infections, *Bom*$^{\Delta55C}$ and *spz*$^{rm7}$ erect wing was taken at 1dpi due to major mortality events by 2dpi specifically in these lines.

Erect wing measurements were performed in parallel with survival experiments, which often introduced injury to the thorax below the wing possibly damaging flight muscle. It is unlikely that muscle damage explains differences in erect wing display. First: we noticed erect wing initially during natural infections with *A. fumigatus*, and observed erect wing upon *B. bassiana R444* and *Metarhizium rileyi PHP1705* natural infections (S1 Table). Second: only 1 of 75 total *iso w*$^{1118}$ males displayed erect wing across 4 systemic infection experiments with *E. faecalis*. For comparison: 19 of 80 total *iso ΔBaraA* and 48 of 80 *w; ΔBaraA* flies displayed erect wing (S1 Table). Future studies might be better served using an abdominal infection mode, which can have different infection dynamics [54]. However we find erect wing display to be robust upon either septic injury or natural infection modes.

## IM10-like peptide in vitro activity

The 23-residue Baramicin peptides were synthesized by GenicBio to a purity of >95%, verified by HPLC. An N-terminal pyroglutamate modification was included based on previous peptidomic descriptions of Baramicins IM10, IM12, and IM13 [55], which we also detected in our LC-MS data (S2 Fig). Peptides were dissolved in DMSO and diluted to a working stock of 1200μM in 0.6% DMSO; the final concentration for incubations was 300μM in 0.15% DMSO. For microbe-killing assays, microbes were allowed to grow to log-growth phase, at which point they were diluted to ~50cells/μL (for *C. albicans* this was OD ≈ 0.01 in our hands).

Two μL of culture (~100 cells), and 1μL water or antibiotic was mixed with 1μL of a 1:1:1 cocktail of IM10, IM12, and IM13 peptides to a final concentration of 300μM total peptides; 1μL of water + DMSO (final concentration = 0.15% DMSO) was used as a negative control. These 4μL microbe-peptide solutions were incubated for 24h at 4˚C. Microbe-peptide cultures were then diluted to a final volume of 100μL and the entire solution was plated on LB agar or BiGGY agar plates. Colonies were counted manually. For combinatorial assays with bacteria, *C. albicans* yeast, and *B. bassiana R444* spores, peptide cocktails were combined with membrane disrupting antimicrobials effective against relevant pathogens beginning at: 10 μM Cecropin A (Sigma), 500μg/mL Ampicillin, or 250μg/mL Pimaricin (commercially available as "Fungin," InVivogen), serially diluted through to 0.1 μM, 0.5μg/mL, and 4μg/mL respectively.

*Beauveria bassiana R444* spores were prepared by dissolving ~30mg of spores in 10mL PBS, and then 4μL microbe-peptide solutions were prepared as described for *C. albicans* followed by incubation for 24h at 4˚C; this spore density was optimal in our hands to produce distinct individual colonies. Then, 4μL PBS was added to each solution and 2μL droplets were plated on malt agar at 25˚C. Colony diameters were measured 4 days after plating by manually analyzing colony diameters in InkScape v0.92. Experimental batches were included as co-variates in one-way ANOVA analysis. The initial dataset approached violating Shapiro-Wilk assumptions of normality (p = 0.061) implemented in R 3.6.3. We subsequently removed four colonies from the analysis, as these outliers had diameters over two standard deviations lower than their respective mean (removed colonies: PBS 15mm, PBS 25mm, IM10-like+Pimaricin 21mm, and a second IM10-like+Pimaricin colony of 21mm); the resulting Shapiro-Wilk p-value = 0.294, and both QQ and residual plots suggested a normal distribution. Final killing activities and colony surface areas were compared by One-way ANOVA with Holm-Sidak multiple test correction (*C. albicans*) and Tukey's honest significant difference multiple test correction (*B. bassiana R444*).

### Gene expression analyses

RNA was extracted using TRIzol according to manufacturer's protocol. cDNA was reverse transcribed using Takara Reverse Transcriptase. qPCR was performed using PowerUP mastermix from Applied Biosystems at 60˚C using primers listed in S3 Data. Gene expression was quantified using the PFAFFL method [56] with *Rp49* as the reference gene. Statistical analysis was performed by one-way ANOVA with Holm-Sidak's multiple test correction or student's t-test. Error bars represent one standard deviation from the mean.

### Proteomic analyses

Raw haemolymph samples were collected from immune-challenged flies for MALDI-TOF proteomic analysis as described in [15,16]. MALDI-TOF proteomic signals were confirmed independently at facilities in both San Diego, USA and Lausanne, CH. In brief, haemolymph was collected by capillary and transferred to 0.1% TFA before addition to acetonitrile universal matrix. Representative spectra are shown. Peaks were identified via corresponding m/z values from previous studies [8,24]. Spectra were visualized using mMass, and figures were additionally prepared using Inkscape v0.92.

### Supporting information

**S1 Fig. Supplemental *BaraA* expression patterns. A)** 400bp of upstream sequence from *BaraA* annotated with putative *Rel* or *Dif/dl* binding sites (included in S1 Data). **B)** Expression of *BaraA in wild-type* and *spz^{rm7}* flies following injury with the Gram-negative bacterium

*E. coli* or the Gram-positive bacterium *M. luteus*. As seen in a previous microarray (Fig 1A), basal *BaraA* expression is depressed in *Rel^E20* flies, but is nevertheless highly induced upon infection, likely representing the *BaraA* response to injury. **C)** In a separate set of experiments, *BaraA* returns to near-baseline levels of expression by 24hpi using *E. coli*. Meanwhile *BaraA* remained induced after pricking with *M. luteus*, mirroring the Toll-regulated *BomBc3* but not the Imd-regulated *DptA*. **D)** The *BaraA>mGFP* reporter line shows a robust induction of GFP 2hpi upon pricking with *M. luteus* in larvae. **E)** Expression of *BaraA>mGFP* in the spermatheca of females (yellow arrow). Representative images shown.
(TIF)

**S2 Fig. LCMS coverage of trypsin-digested and detected BaraA peptides aligned to the protein coding sequence.** Detected peptide fragments (blue bars) cover the whole precursor protein barring furin site-associated motifs. Additionally, two peptide fragments are absent: i) the first 4 residues of the C-terminus ("GIND," not predicted *a priori*), and ii) the C-terminus peptide's "RPDGR" motif, which is predicted as a degradation product of Trypsin cleavage and whose size is beyond the minimum range of detection. Without the GIND motif, the mass of the contiguous C-terminus is 5974.5 Da, matching the mass observed by MALDI-TOF for IM22 (Fig 2A). The N-terminal Q residues of IM10, IM12, IM13, and IM24 are pyroglutamate-modified, as described previously [24]. The asparagine residues of IM10-like peptides are sometimes deamidated, likely as a consequence of our 0.1% TFA sample collection method as "NG" motifs are deamidated in acidic conditions [58].
(TIF)

**S3 Fig. Alignments of BaraA peptide motifs. A)** Aligned IM22 peptides of *Drosophila Baramicin A-like* genes, with the IM10-like 'VWKRPDGRTV' motif noted. The GIND residues at the N-terminus are cleaved off in *Dmel\BaraA* by an unknown process, and this subsequent peptide is similarly cleaved following RXRR furin cleavage sites in subgenus Drosophila flies. As a consequence, the mature IM22 peptide is predicted to be the same across species even when different cleavage mechanisms are utilized. **B)** Alignment of the three IM10-like peptides of *D. melanogaster BaraA* with the "VXRPXRTV" motif noted. The residue 8 polymorphism of either G (IM12) or D (IM10, IM13) has evolved repeatedly in outgroup flies [25], indicating it is likely key for IM10-like peptide activity.
(TIF)

**S4 Fig. Over-expression of *BaraA* partially rescues *Rel*, *spz* double mutant susceptibility to infection in both males and females. A)** Validation of the *UAS-BaraA* construct in the *Rel*, *spz* background. Flies were unchallenged. **B)** Overexpressing *BaraA* did not improve the survival of *Rel*, *spz* flies upon *E. coli* infection. **C)** Overexpressing *BaraA* only marginally improves survival of *Rel*, *spz* females, but not males, upon *M. luteus* infections. Infections using a higher dose (OD = 100) tended to kill 100% of *Rel*, *spz* flies regardless of sex or expression of *BaraA*, suggesting that if *BaraA* overexpression does affect susceptibility to *M. luteus*, this effect is possible within only a narrow window of *M. luteus* concentration. **D-F)** Overexpressing *BaraA* improves survival of *Rel*, *spz* male and female flies upon injury with *C. albicans* (**D**) or natural infection with *A. fumigatus* (**E**) and *N. crassa* (**F**). P-values are shown for each biological sex in an independent CoxPH model not including the other sex relative to *Rel*, *spz* as a reference.
(TIF)

**S5 Fig. RT-qPCR shows that the expression of *BomBc3* (A) *Drs* (B) and *DptA* (C) is wild-type 18hpi in *iso ΔBaraA* flies. D)** *BaraA* mutants survive clean injury like wild-type flies.

**E)** *iso ΔBaraA* flies have similar lifespan compared with the *iso w^{1118}* wild-type (males + females, *iso* vs. *iso ΔBaraA*: HR = 0.26, p = .118)
(TIF)

**S6 Fig. Additional survivals using *ΔBaraA* flies in two distinct genetic backgrounds upon infection by a diversity of microbes. A-B)** No significant susceptibility of *ΔBaraA* flies to *Ecc15* **(A)**, *P. burhodogranariea* **(B)**, or *B. subtilis* **(C)**, bacterial infections. **D-E)** *w; ΔBaraA* males were slightly susceptible to *A. fumigatus* natural infection (HR > 0.5, p = .078), but not females, nor isogenic flies. Additional infections using *ΔBaraA, Bom^{Δ55C}* double mutant flies reveals that *BaraA* mutation increases the susceptibility of *Bom^{Δ55C}* flies in both males and females (cumulative curves shown in Fig 5A). Blue backgrounds = Gram-negative bacteria, orange backgrounds = Gram-positive bacteria, yellow backgrounds = fungi.
(TIF)

**S7 Fig. Survival analysis suggests a minor contribution of *BaraA* to defence against infection by *E. faecalis*. A)** *w; ΔBaraA* but not *iso ΔBaraA* flies are significantly susceptible to *E. faecalis*. However we note that *iso ΔBaraA* flies suffer an earlier mortality than *iso w^{1118}* wild-type controls that is highly significant if the experiment is artificially censored at 3.5 days (dotted line and associated statistics). **B)** Crosses with a genomic deficiency (*Df(BaraA)*) leads to increased susceptibility in both the *w* background and isogenic DrosDel background, with *Df(BaraA)/ΔBaraA* flies suffering the greatest mortality in either crossing scheme. Both deficiency crosses yielded an earlier susceptibility in *BaraA*-deficient flies (shown with dotted black lines), however neither experiment ultimately reached statistical significance. **C)** *BaraA* RNAi flies (*Act>BaraA-IR*) suffered greater mortality than *Act>OR-R* or *OR-R/BaraA-IR* controls, but this was not statistically significant at α = .05; p-values reported are comparisons to *Act>BaraA-IR* flies.
(TIF)

**S8 Fig. Additional survival analyses reveal a consistent contribution of *BaraA* to defence against infection by *B. bassiana*. A)** *BaraA* mutants in both backgrounds are highly susceptible to natural infection with the entomopathogenic fungus *B. bassiana 802*. **B)** Crossing with a genomic deficiency (*Df(BaraA)*) leads to increased susceptibility of *Df(BaraA)/ΔBaraA* flies for both the *w* background and isogenic DrosDel background relative to wild-type controls (p < .05) upon *B. bassiana 802* natural infection. **C)** *Act>BaraA-IR* flies were more susceptible than the *OR-R* wild-type (p = .008) and *OR>BaraA-IR* (p = .004), although not significantly different from our *Act>OR-R* control (p = .266). **D)** Overexpressing *BaraA* (*Act>UAS-BaraA*) improved survival against *B. bassiana 802* relative to *Act>OR-R* controls (HR = -0.52, p = 0.010). **E)** *BaraA* alone contributes to survival against *B. bassiana* to a far greater extent than the two canonical antifungal peptide genes *Mtk* and *Drs*, which in fact had little effect on survival outcome.
(TIF)

**S9 Fig. Frequency of erect wing display following additional challenges. A)** Erect wing occurs in flies given natural infection with *A. fumigatus*, wherein flies do not readily succumb to infection (S6D Fig) and no thoracic injury was introduced. **B-C)** Erect wing frequencies 2dpi after clean injury **(B)**, or *Ecc15* septic injury **(C)**. The erect wing frequencies of flies pricked by HK-*E. faecalis* (Fig 6C) are included in brown to facilitate direct comparison with the frequency observed upon Toll pathway activation. **D)** The frequency of erect wing display is increased following *E. faecalis* septic injury in *ΔBaraA/+* or *Df(BaraA)/+* flies. Data points are pooled from *w; ΔBaraA* and *iso ΔBaraA* crosses after *E. faecalis* infections shown in S7A Fig and data in S1 Table. **E)** $C_T BaraA-C_T Rp49$ (*ΔC_T*) non-normalized expression of the

*BaraA-Gal4>UAS-BaraA* method to better visualize expression level differences. This Gal4/UAS approach rescues *BaraA* expression in *ΔBaraA* flies, though not quite to wild-type levels. A very low level of expression was observed in *ΔBaraA, UAS-BaraA/ΔBaraA* flies without the Gal4 (indicating a tiny level of UAS leakiness), while *BaraA* was never detected in *w; ΔBaraA* flies. Differences in this $\Delta C_T$ y-axis effectively equate to Log2 expression differences. The level of *BaraA* induction in these *ΔBaraA, BaraA-Gal4>UAS-BaraA* was ~3.3x the unchallenged state by 24hpi.
(TIF)

**S10 Fig. *ΔBaraA*/+ transheterozygotes suffer significantly reduced *BaraA* expression. A)** Schematic detailing the *BaraA* loci of genotypes used in transheterozygote crosses. **B-C)** *BaraA* **(B)** and *BomBc3* **(C)** expression after *B. bassiana* pricking in *BaraA* homozygous or heterozygous flies. Transheterozygotes with one mutant locus have significantly reduced *BaraA* expression. Intriguingly, *OR-R* flies (homozygous for 2 gene copies) have higher *BaraA* expression levels compared to $w^{1118}$ (1 gene copy) after infection **(B)**, which appears to be unrelated to the activation of the Toll response generally as *BomBc3* levels were comparable across genotypes **(C)**. Instead, *OR-R* flies seemingly reach a slightly greater absolute expression (S9E Fig). Statistically significant differences at 24hpi are indicated by red letters, to facilitate complex multiple comparisons (one-way ANOVA with Holm-Sidak's multiple test correction). Genotypes with the same letter group are not significantly different from each other. In all cases, no significant differences were observed amongst unchallenged flies.
(TIF)

**S1 Table. Erect wing frequencies from various infection experiments.** Following initial erect wing observations after *A. fumigatus* natural infection, we scored erect wing frequency in all subsequent survival experiments. Data represent the mean % of males displaying erect wing ± one standard deviation. n exp = number of replicate experiments performed, and dpi ewg taken = days post-infection where erect wing data were recorded. We additionally performed natural infections with *Metarhizium rileyi* that generally did not cause significant mortality even in *ΔBaraA, $Bom^{Δ55C}$* double mutant males, but nevertheless induced erect wing specifically in *ΔBaraA* males and *$spz^{rm7}$* controls. Bacterial infections were performed by septic injury, while fungal challenges were either natural infections (NI) performed by rolling flies in spores or septic injuries as indicated. Underlying data are included in S5 Data.
(XLSX)

**S1 Text. Supplementary discussion of IM22 identification and *BaraA-Gal4* construct.**
(DOCX)

**S1 Data. Putative NF-κB sites in the *Baramicin* promoter.**
(XLSX)

**S2 Data. Standard curves to calculate peptide masses in Uttenweiler-Joseph et al. [8] and this study to identify IM22, and charge characteristics of Baramicin peptides.**
(XLSX)

**S3 Data. Primers used in this study and annotation of BaraA copy number in a selection of wild-type strains.**
(XLSX)

**S4 Data. Fly stock and Microbe strain information.**
(XLSX)

**S5 Data. Complete erect wing data S1 Table.**
(XLSX)

## Acknowledgments

We thank Jean-Philippe Boquete for assistance with the generation of Gal4 and UAS constructs. We would also like to acknowledge the technical expertise provided by the proteomics and mass spectrometry facilities in both UCSD and EPFL, and specifically Adrien Schmid. The name "*Baramicin*" was partly inspired by Eichero Oda's character "Buggy," a Bara-Bara superhuman. Finally, we further thank Dominique Ferrandon and Jianqiong Huang et al. [57] for their cooperation in publishing initial descriptions of the *BaraA* gene, and for stimulating discussion.

## Author Contributions

**Conceptualization:** Mark Austin Hanson, Steven A. Wasserman.

**Data curation:** Mark Austin Hanson.

**Formal analysis:** Mark Austin Hanson.

**Funding acquisition:** Steven A. Wasserman, Bruno Lemaitre.

**Investigation:** Mark Austin Hanson, Lianne B. Cohen, Alice Marra.

**Methodology:** Mark Austin Hanson, Lianne B. Cohen, Igor Iatsenko, Steven A. Wasserman, Bruno Lemaitre.

**Project administration:** Bruno Lemaitre.

**Resources:** Mark Austin Hanson, Lianne B. Cohen, Igor Iatsenko, Steven A. Wasserman, Bruno Lemaitre.

**Supervision:** Steven A. Wasserman, Bruno Lemaitre.

**Validation:** Mark Austin Hanson, Lianne B. Cohen.

**Visualization:** Mark Austin Hanson.

**Writing – original draft:** Mark Austin Hanson.

**Writing – review & editing:** Mark Austin Hanson, Bruno Lemaitre.

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
