## [Decision Letter · Decision Letter 0]

22 Jul 2021

Dear Prof. Lemaitre,

Thank you very much for submitting your manuscript "The Drosophila Baramicin polypeptide gene protects against fungal infection" for consideration at PLOS Pathogens. As with all papers reviewed by the journal, your manuscript was reviewed by members of the editorial board and by several independent reviewers. The reviewers appreciated the attention to an important topic. Based on the reviews, we are likely to accept this manuscript for publication, providing that you modify the manuscript according to the review recommendations.

Sincerely,

Xiaorong Lin, Ph.D.

Section Editor

PLOS Pathogens

Xiaorong Lin

Section Editor

PLOS Pathogens

Kasturi Haldar

Editor-in-Chief

PLOS Pathogens

orcid.org/0000-0001-5065-158X

Michael Malim

Editor-in-Chief

PLOS Pathogens

orcid.org/0000-0002-7699-2064

Reviewer Comments (if any, and for reference):

Reviewer's Responses to Questions

**Part I - Summary**

Reviewer #1: I have already reviewed this manuscript once, as part of the Review Commons -system (as Reviewer #3). I feel that my concerns / questions / suggestions at that stage have been adequately addressed and I don't have any more suggestions.

Reviewer #2: The authors use the fruitfly Drosophila melanogaster as a model to study innate immunity. In this manuscript, they study the effects of a set of antimicrobial peptides (AMPs) that are produced by furin cleavage of a larger precursor (Baramicin A, BaraA). Bara A is immune-induced in a Toll-dependent manner and has antifungal activity. Somewhat in line with expression in non-immune tissues, BaraA mutants show ab erect-wing phenotype in males.

Reviewer #3: BaraA is a Toll-dependent gene expressed in the fat body upon immune challenge that codes for a protein that is proteolytically processed into 8 smaller peptides. Additionally, the BaraA locus is frequently duplicated in several lines of D. melanogaster and this gene is expressed in other tissues (head, eyes, etc.) at baseline. Male BaraA mutants displayed an erected-wings phenotype when subjected to infection, which can be rescued through overexpression of BaraA in the fat body.

There are 3 key findings:

- BaraA overexpression conferred protection against fungal infection.

- BaraA-derived peptides displayed antifungal activity in conjunction with Pimaricin in vitro.

- Loss of BaraA decreased fungal resistance, independent of the Bomanins.

**Part II – Major Issues: Key Experiments Required for Acceptance**

Reviewer #1: (No Response)

Reviewer #2: (No Response)

Reviewer #3: The authors addressed and resolved many issues in previously draft. The additional results were clear and the use of various backgrounds was justified. However Lines 140-148 are very confusing. The authors claim BaraA is highly induced in the fat body by M. luteus but not E. coli, and triggered by the Toll pathway and not the Imd pathway , but S1B showed BaraA being induced strongly (45x) at 6hpi by E. coli, and lesser extent (6x) by M. luteus at 24hpi. And while the E. coli triggered expression is mostly independent of Relish, this experimental design does not rule out other components of the Imd pathway, nor did the data and the use of reference definitively prove that this is triggered by Toll in response to E. coli. The methodology of figure 1C, which indicates a more robust response to M. luteus (but not unresponsive to E. coli - the use of “not” in the text and figure legend is unjustified) is not explained in either the text or the legend. Further, the analysis in figure 1C appears to be only a binary call, based on a reporter over a timecourse of tissue localization, but does not consider the intensity, making it difficult to make quantitative conclusions on the inducibility of this gene by one bacteria over the other. Overall, the information presented is very disjointed; it is difficult to arrive at the authors’ conclustion that the expression of BaraA is strongly induced in a Toll-dependent manner in the fat body with the combination of the figures presented (1C & S1B)], but unresponsive to E. coli and the Imd pathway. The authors should be more thorough with their analysis or more judicious with their conclusions.

The first four residues of the C-terminal in D. melanogaster BaraA were not mentioned in the text. They were identified as GIND motif in the figure 1B legend, different from a furin cleavage motif (RXRR) at this site in other Drosophila species. The figure 2B itself, however, did not make this distinction clear and denoted this site as a furin cleavage site similar to other sites in D. melanogaster. Is this GIND motif a cleavage site for furin as well? Or is it cleaved by something else? Is there a significance in this site being different in D. melanogaster vs other Drosophila species?

**Part III – Minor Issues: Editorial and Data Presentation Modifications**

Reviewer #1: I found two typos or spelling mistakes:

In the abstract (line 23) Drososphila melanogaster -> Drosophila melanogaster

Line 134: ...flies similar tolike the Toll-regulated BomBc3 -> ...flies similar to the Toll-regulated BomBc3

Reviewer #2: (No Response)

Reviewer #3: Does the duplication event affect expression? It would be nice to have a brief comment about this in this paper regardless of the content of the other preprint.

The diagram depicting the deletion in figure 2A is hard to read. It is too small and contains too little detail. It also looks as if the authors were trying to line the deletion diagram up with the MALDI-TOF graph, but these two graphs are not related to one another.

PLOS authors have the option to publish the peer review history of their article (what does this mean?). If published, this will include your full peer review and any attached files.

Reviewer #1: No

Reviewer #2: No

Reviewer #3: No

Figure Files:

Data Requirements:

Reproducibility:

References:

---

## [Editor Report · Decision Letter 1]

28 Jul 2021

Dear Prof. Lemaitre,

We are pleased to inform you that your manuscript 'The Drosophila Baramicin polypeptide gene protects against fungal infection' has been provisionally accepted for publication in PLOS Pathogens.

Best regards,

Xiaorong Lin, Ph.D.

Section Editor

PLOS Pathogens

Xiaorong Lin

Section Editor

PLOS Pathogens

Kasturi Haldar

Editor-in-Chief

PLOS Pathogens

orcid.org/0000-0001-5065-158X

Michael Malim

Editor-in-Chief

PLOS Pathogens

orcid.org/0000-0002-7699-2064
---

## [Editor Report · Acceptance letter]

11 Aug 2021

Dear Prof. Lemaitre,

We are delighted to inform you that your manuscript, "The Drosophila Baramicin polypeptide gene protects against fungal infection," has been formally accepted for publication in PLOS Pathogens.

Best regards,

Kasturi Haldar

Editor-in-Chief

PLOS Pathogens

orcid.org/0000-0001-5065-158X

Michael Malim

Editor-in-Chief

PLOS Pathogens

orcid.org/0000-0002-7699-2064